# Foraging activity of harbour porpoises around a bottom-gillnet in a coastal fishing ground, under the risk of bycatch

Saki Maeda[1], Kenji Sakurai[2], Tomonari Akamatsu[3], Ayaka Matsuda[4], Orio Yamamura[4], Mari Kobayashi[5], Takashi Fritz Matsuishi[4]*

1 Graduate School of Fisheries Sciences, Hokkaido University, Hakodate, Hokkaido, Japan, 2 Rausu Fisheries Cooperative Association, Rausu-cho, Hokkaido, Japan, 3 Ocean Policy Research Institute, Sasakawa Peace Foundation, Minato-ku, Tokyo, Japan, 4 Faculty of Fisheries Sciences, Hokkaido University, Hakodate, Hokkaido, Japan, 5 Faculty of Bioindustry, Tokyo University of Agriculture, Abashiri, Hokkaido, Japan

* catm@fish.hokudai.ac.jp

**Data Availability Statement:** All relevant data are within the manuscript.

## Abstract

Bycatch of harbour porpoises (*Phocoena phocoena*) by gillnets is a recognised threat to populations. To develop effective mitigation measures, understanding the mechanics of bycatch is essential. Previous studies in experimental conditions suggested foraging activity is an important factor influencing porpoises' reaction to gillnets. We acoustically observed the behaviour of wild harbour porpoises around a bottom-gillnet set-up in a commercial fishing ground, especially foraging activity. Passive acoustic event recorders (A-tags) were fixed to the ends of the gillnet, and recorded for 1 392 hours. Although harbour porpoises frequently and repeatedly appeared around the net each day, incidental bycatch occurred only three times during the observations. The stomach contents of two individuals contained mainly *Ammodytes* sp., which were observable around the bottom-gillnet but not targeted by the fishery. A total of 276 foraging incidents were acoustically detected, and 78.2% of the foraging activity was in the bottom layer (deeper than 25 m). Porpoises appeared around the net with more frequency on the day of a bycatch incident than on the days without bycatch. These results suggest that the harbour porpoises appeared around the bottom-gillnet to forage on fish distributed in the fishing ground, but not captured by this bottom-gillnet. Thus, porpoises face the risk of becoming entangled when foraging near a gillnet, with the probability of bycatch simply increasing with the length of time spent near the net. Bycatch mitigation measures are discussed.

## Introduction

Gillnets are a passive fishing gear that entangle animals in the mesh, and are operated worldwide based on their versatilely and fuel efficiency [1, 2]. Bycatch in gillnets has been increasingly recognised as a significant threat to animal populations, including seabirds, sea turtles and marine mammals [3–5]. It is confirmed that gillnet bycatch affects the population sustainability of some small cetaceans [6]; especially, the vaquita (*Phocoena sinus*) is in danger of

**Funding:** The Ministry of Education, Culture, Sports, Science and Technology, Japan (MEXT) to a project on Joint Usage/Research Center– Leading Academia in Marine and Environmet Pollution Research (LaMer) http://lamer-cmes.jp/ MATSUISHI Takashi Fritz JSPS KAKENHI Grant Number 26450255 https://kaken.nii.ac.jp/ja/grant/KAKENHI-PROJECT-26450255/ (MATSUISHI Takashi Fritz, AKAMATSU Tomonari) JSPS KAKENHI Grant Number 18J30013 https://kaken.nii.ac.jp/grant/KAKENHI-PROJECT-18J30013/ (MATSUDA Ayaka) For all the funds above, the funders did not play a role in the study design, data collection and analysis, decision to publish, nor preparation of the manuscript.

**Competing interests:** NO authors have competing interests

extinction because of bycatch [7]. Bycatch of small cetacean may have merit to the fishers in many parts of the world (e. g. [8, 9]) for consumption or for use as bait in gillnet and longline fisheries [10]. On the other hand, it can also disturb fishing operations [11]: although they are considered 'small' cetaceans, most of the porpoises exceed 1 m in body length; when bycatch occurs, fishers face difficulty removing them from nets.

The harbour porpoise (*Phocoena phocoena*) is one species that is incidentally caught by gillnets. Because their habitat is close to coastlines, they are subject to incidental capture in gillnets throughout their habitat, including in fishery waters off Canada, United States, Japan, Scotland, France, Germany, Sweden, Poland, Denmark, Norway, Iceland and Greenland [12]. Populations of harbour porpoise are threatened in some areas [6, 13–15]. To reduce the bycatch of this species, mitigation measures have been attempted both in Europe and North America (e.g. [16–18]), such as time–area regulations of a fishery, technological modifications of the fishing gear, and use of acoustic deterrent devices called pingers. Despite these efforts, harbour porpoise bycatch still occurs [19]; for example, even though pingers are temporarily effective in reducing bycatch, the widespread use of them with gillnets would likely be insufficient to eliminate porpoise bycatch [20], as their effectiveness remains a subject of dispute [21–24].

In Japan, almost 25% of fishing operators use gillnets, mostly in a small-scale gillnet fishery (vessel size of up to 5 gross tons). Especially in Hokkaido, gillnets are the most-operated fishing gear, used by more than 3 000 fishing operator households [25]. Around Japan, harbour porpoises are distributed mainly nearshore off Hokkaido [26–29]. The Stranding Network Hokkaido (SNH), a local stranding network organised by scientists, museum curators, and fishers managed by the corresponding author M.T.F., has been collecting information on gill net bycatch along the Hokkaido coast. According to the stranding reports disclosed from SNH (https://kujira110.com), one gillnet fisher (S. K., a co-author) reported up to 10 harbour porpoises bycatches occurred in his bottom-gillnets up to 10 times year$^{-1}$. Although the total number of bycatch and incidence around Hokkaido remain unknown, because there is no obligation to report the total number of bycatch incidences. It is assumed that a substantial number of harbour porpoises are taken as bycatch around Hokkaido, drowning in gillnets could be one of the most significant threats for harbour porpoises in the Hokkaido area.

To develop effective and long-lasting mitigation measures, an understanding of the bycatch mechanism is essential. For instance, knowledge of the behaviour of harbour porpoises around nets helps to determine the conditions contributing to the incidence of bycatch [12, 30]. In the case of bottlenose dolphins (*Tursiops truncatus*), there are reports of depredation [31], as well as observations of damage to captured fish or bait, presumably caused by the dolphins [32]. Other researchers showed that bottlenose dolphins spend time foraging around gillnets [33]. However, why harbour porpoises appear around fishing nets and become entangled is not entirely clear [34].

To the best of our knowledge, no study that directly observed foraging activity of harbour porpoise around gillnet is reported. A study [35] just suggested the possibility that harbour porpoises had been feeding just before becoming entangled because the stomach contents of individuals in the fishing net included intact herring. Alternately, another study [36] suggested that harbour porpoises will not approach gillnets when foraging. Thus far, no study has presented clear evidence that harbour porpoises prey around gillnets.

An effective way to observe the foraging activity of harbour porpoises would be an acoustical survey, as visual observations underwater are relatively difficult. Previous studies succeeded in observing the foraging activity of harbour porpoise by recording their echolocation sounds or clicks [37]. Harbour porpoises are known to echolocate almost continuously [38], for the sake of orientation and prey capture [39], and they emit specific clicks patterns while foraging [40]. Thus, recordings of the click patterns allow us to classify their behaviours [37, 41].

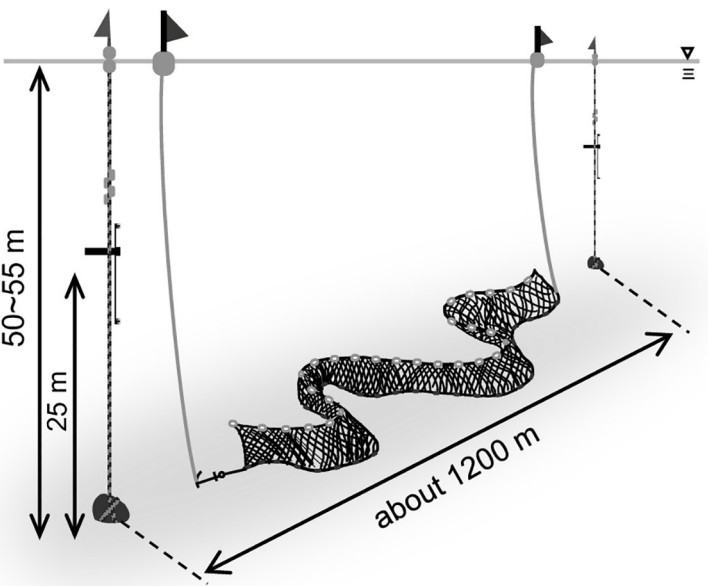

**Fig 1. Illustration of the A-tag and bottom-gillnet set-up monitored off Rausu, Hokkaido, Japan.**

To mitigate porpoise bycatches, fishers must be good observers [12], especially around gill-nets. An author (K.S.) a member of the SNH who operates a bottom-gillnet fishery in Hokkaido, deployed acoustic data-loggers called A-Tag beside the net (Fig 1) for 1 392 hours in 58 days (Table 1), and has reported all bycatches of harbour porpoises (Fig 2, Table 2); which enabled acoustic monitoring near the net, especially immediately before an individual porpoise was caught. From the record of A-Tag, we observed the daily appearance patterns, foraging behaviours, appearance and foraging depth.

The purpose of this study was to clarify the behaviour of harbour porpoises around a bottom-gillnet set in a commercial fishing ground, off Hokkaido, Japan, by passive acoustic monitoring.

## Results

### Characteristic of harbour porpoise presence and foraging activity

A total of 10 313 possible clicks were identified, after the screening of the clicks. Clicks were detected every day during the observations. A total of 520 presences were counted, grouping the clicks by a presence threshold interval (PTI) described in the Materials and Methods section. A frequency plot of all the presences for each hour (Fig 3) showed a bell-shaped curve, with a peak at 23:00.

The average (±S.D.) of presence duration for overall presence was calculated as 534 ± 1 043 sec. The duration exceeded one hour for 12 of the presences, whereas the most continuous presence lasted for 7 505 sec.

**Table 1. Periods of the A-tag acoustic observations and the bycaught harbour porpoises.**

| Year | Start | End | Days | No. of bycatches |
|------|-------|-----|------|------------------|
| 2016 | 9 Jul | 25 Jul | 16 | 1 |
| 2017 | 7 Jul | 29 Jul | 22 | 2 |
| 2018 | 21 Jul | 10 Aug | 20 | 0 |
| Total | | | 58 | 3 |

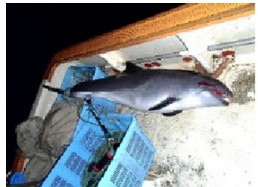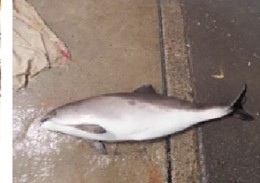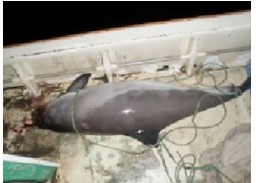

**Fig 2. Photographs of the three bycaught harbour porpoises reported on here (see also Table 2).**

The histogram in elevation/depression angle of all obtained pulses is shown in Fig 4. More than half the pulses (59.6%) had a negative value for the elevation/depression angle. As stated below, the elevation/depression angle was used as an index of the swimming depth-layer of the harbour porpoises, and a negative value for the angle indicated that the click came from below (deeper than 25-m bottom layer) the A-tag. This means that more than half the recorded clicks emanated from the direction of the seafloor.

Foraging sequences were observed 276 times; a typical example of foraging sequence is shown in Fig 5. According to the negative value in this instance, this foraging sequence was determined to be emitted from the direction of the seafloor, and it includes the approach phase (both the initial and terminal parts).

The observed foraging sequences showed a diel pattern (Fig 6), being more frequent during the night-time (19:00–5:00; 75.4%) than the daytime (5:00–19:00). The average number of foraging sequences hour$^{-1}$ in night-time (20.8) was significantly higher than that in daytime (5.2, $t$-test, $p < 0.05$).

## Characteristics of behaviours before bycatch

Presence and foraging activity were compared between days with and without a bycatch event (Table 3). The average number of presences before bycatch period (BB) (21.3) was significantly greater than for no bycatch period (NB) (8.4, $t$-test, $p < 0.001$). Furthermore, the average presence probability for BB (19.2%) was also significantly higher than for NB (4.7%, $t$-test, $p < 0.001$). However, the other variables, such as presence duration, elevation/depression angle, and the number of foraging sequences, showed no significant differences between BB and NB (Table 3).

The histogram of the elevation/depression angle of all foraging sequences is shown in Fig 4. Most foraging sequences (78.9%) were observed from the sea-bottom direction.

## Stomach-contents analysis

Food remains were found in the stomachs of both porpoises that were bycatch. The total stomach contents included 301 otoliths; of these, 294 otoliths (94.3%) were identified as from *Ammodytes* sp., and the others could not be identified because of an advanced stage of digestion (Table 4). One cephalopod lower beak was found in the stomach of SNH17205, and was

**Table 2. Summary of harbour porpoise bycatch in the gillnet set off Rausu, Hokkaido, Japan, during the observation period.** The stomach contents of specimen SNH16208 were not available.

| Sample number | Bycatch date | Sex | Body length (cm) | Stomach contents |
|---|---|---|---|---|
| SNH16208 | 25 Jun 2016 | female | 126.0 | — |
| SNH17204 | 10 Jul 2017 | male | 143.5 | Table 4 |
| SNH17205 | 29 Jul 2017 | female | 161.0 | Table 4 |

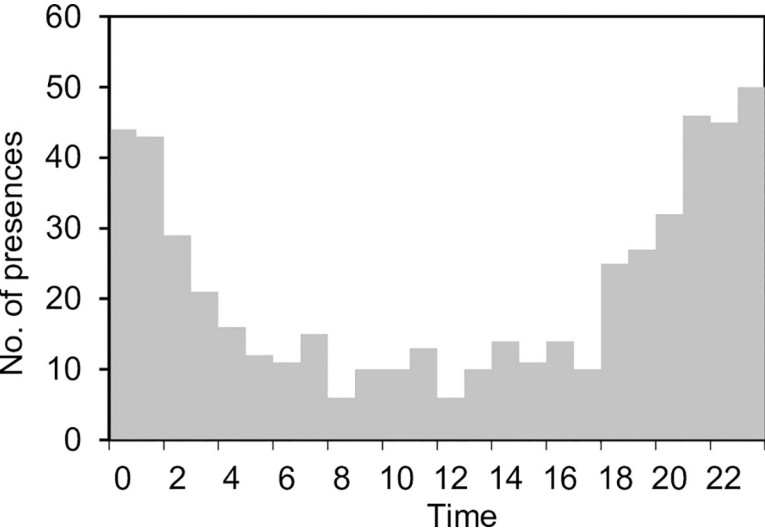

**Fig 3. Number of harbour porpoise present around the bottom-gillnet, as detected by the A-tag, for different hours of the day.**

an unidentified *Octopotidae* sp. In addition, fish bones and eyeballs and parasitic worms were also found, but were not further identifiable to taxonomic group.

## Discussion

To our knowledge, the current study is the first attempt to observe the foraging activity of harbour porpoises around a bottom-gillnet as an actual fishery deployment. Harbour porpoises appeared around the bottom-gillnet set-up every day during the observation periods, and the vast majority of instances of presence around the net did not result in bycatch. However, for the period category BB (on the day prior to a set with a bycatch event), the number of presences and the presence probability were greater than for the period category NB (days prior to sets with no bycatch event). The average of presence duration was about 9 min, and, in some

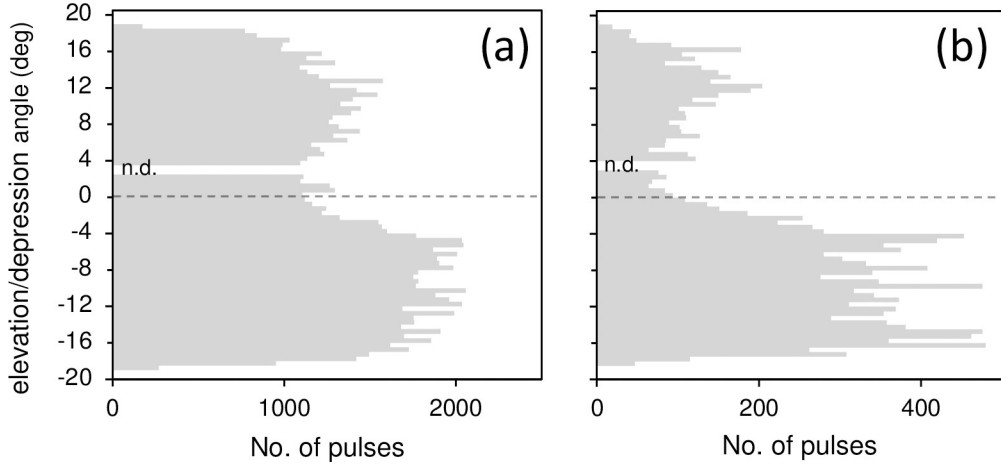

**Fig 4.** Frequency distributions of elevation/depression angle for (a) all pulses recorded and (b) the foraging sequences for harbour porpoises by the A-tag in the gillnet set-up during the present study. Pulses with an elevation/depression angle between 2.5˚ and 3.5˚ were omitted.

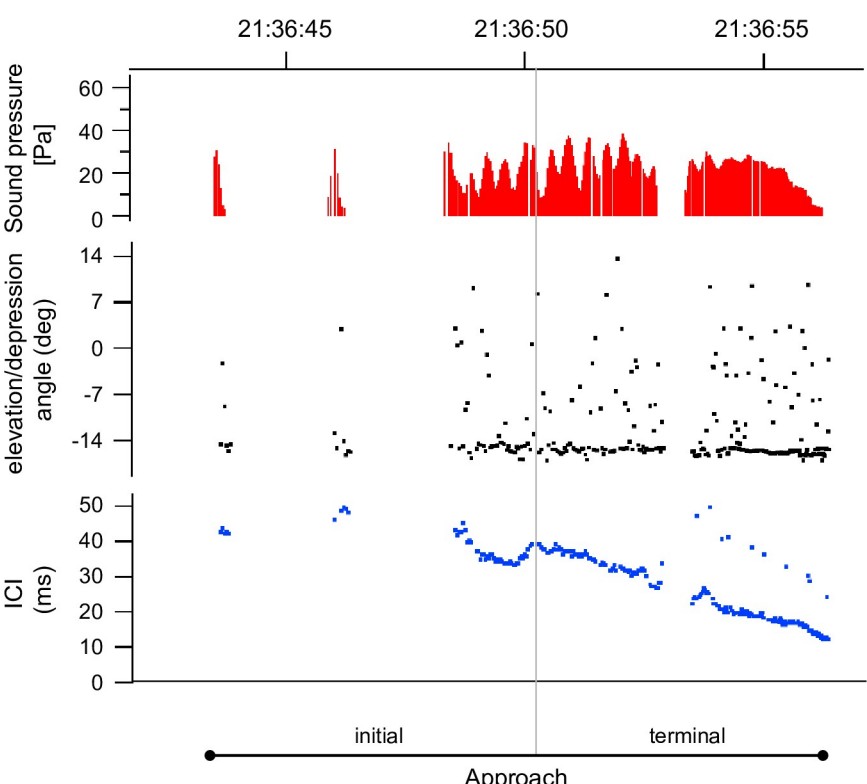

**Fig 5. An example of foraging sequences recorded around the bottom-gillnet on 22 July 2016.** Time-series for sound pressure, elevation/depression angle, and inter-click interval are shown.

extreme cases, the presence duration exceeded 1 h—too long for a porpoise to be just passing along the net. Based on the results, it is suggested that harbour porpoises often stayed around the net, rather than just appearing as they passed by, even when no bycatch occurred.

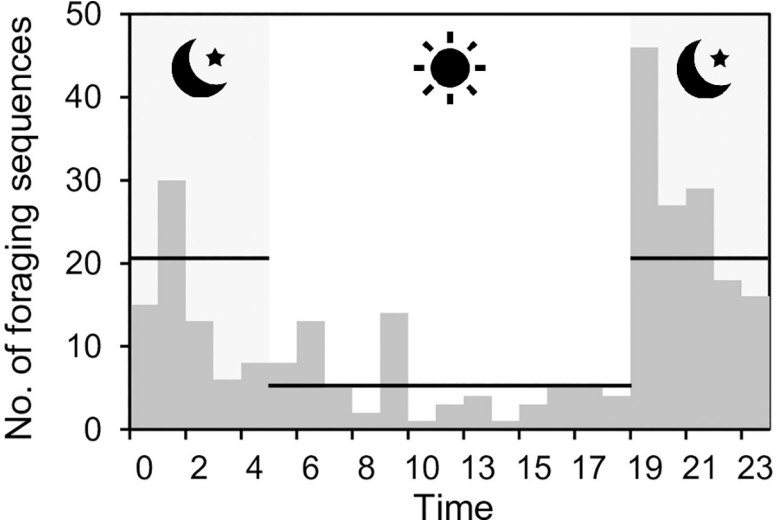

**Fig 6. Diel pattern of harbour porpoise foraging sequences around the bottom-gillnet recorded in the present study.** Horizontal lines indicate mean number of foraging sequences hour$^{-1}$, for night-time (20.8 ± 3.8, mean ± S.E.) and daytime (5.2 ± 1.1).

**Table 3. Comparison of harbour porpoise behaviour variables between the 'before bycatch (BB)' and 'no bycatch (NB)' periods (mean ± SE).**

|  | Before bycatch | | No bycatch | | p (*t*-test) |
|---|---|---|---|---|---|
| No. of presences | 21.3 ± | 6.5 | 8.4 ± | 0.5 | <0.001 |
| Presence duration (sec) | 658.0 ± | 221.0 | 516.0 ± | 46.0 | ns |
| Presence probability (%) | 19.2 ± | 8.7 | 4.7 ± | 1.0 | <0.001 |
| Elevation/depression angle (degrees) | −2.0 ± | 0.2 | −2.0 ± | 0.1 | ns |
| No. of foraging sequences | 1.7 ± | 0.7 | 4.3 ± | 1.2 | ns |

ns: not significant

Harbour porpoises dive and surface repeatedly, and more than 85% of dives may be shallower than 20 m [42, 43]. However, in the present study, more than half of the recorded clicks (59.3%) were emitted deeper than 25 m. Compared with the general vertical distribution of wild harbour porpoises, the observed presence of porpoises around the bottom-gillnet set-up had a biased distribution toward the seafloor (Fig 4). Bottom dives (>20 m) by harbour porpoises were associated with foraging [43, 44]. In this study, foraging behaviour was observed most often in the deeper layer (>25 m; 78.2%). Therefore, it is reasonable to suggest that the occurrence of harbour porpoises around the bottom-gillnet was largely related to foraging.

From the stomach-contents analysis, bycatch individuals preyed almost exclusively on *Ammodytes* sp. This species is the most common prey of harbour porpoise round Hokkaido [28] These fish were frequently observed around the site where the net was operated. According to the local fishers, righteye flounders move ashore after sunset; moreover, righteye flounders and sculpins feed on *Ammodytes* sp. just before they are captured in the bottom-gillnet because intact *Ammodytes* sp. are often found in the mouths of these fishes.

Prey distribution often affects the movements or presence of porpoises [45]. Harbour porpoises are known to be highly adaptive and opportunistic in their foraging ecology [41], and their movements and presence are affected by prey fish distributions [45]. It is natural that the distribution of harbour porpoises in this sea area is also affected by prey distribution. Although no direct evidence is available, the occurrence of porpoises around the net was probably related to the distribution of forage fishes, including *Ammodytes* sp.

Although harbour porpoises preyed on fish around the bottom-gillnet, this does not indicate that the porpoises' foraging activity around the net was depredation. *Ammodytes* sp. was not caught in the bottom-gillnet because their body height (<3 cm) is much smaller than the mesh size (7.5 cm). Therefore, it can be concluded that while harbour porpoises foraged around the net, their proximity to the net was not related to depredation. Since *Ammodytes* sp. is abundant and the targeted species of the bottom-gillnet including righteye flounders and sculpins also feed on *Ammodytes* sp. there, this location is a good fishing ground. It seems that their feeding place and the fishery ground are simply overlapping because of the distribution of *Ammodytes* sp.

**Table 4. Stomach contents of two harbour porpoises that became entangled in a bottom gillnet off Rausu, Hokkaido, Japan.**

| Species name | SNH17204 | SNH17205 | Total | $N_i$% |
|---|---|---|---|---|
| OCTOPODA |  |  |  |  |
| *Octopodidae* sp. |  | 1 | 1 | 0.7 |
| TRACHINIFORMES |  |  |  |  |
| *Ammodytes* sp. | 84 | 49 | 133 | 94.3 |
| Unidentified fishes | 4 | 3 | 7 | 5.0 |

From these observations, the occurrences of bycatch can be suggested as incidental. A comparison of the BB and NB periods clarified that harbour porpoises appeared more intensively just before an incidence of bycatch. No difference in elevation/depression angle was observed between the BB and NB periods, suggesting that porpoises did not become entangled because they had dived deeper than in usual. These observations indicate that harbour porpoises risk entanglement by appearing around a bottom-gillnet, and the risk of bycatch increases with repeated appearances, which is similar to the recent report on Peruvian drift net fishery [46].

Calculation of a hazard ratio (HR) [47] provides a method for objectively evaluating the relative risk of bycatch. HR is often defined as the ratio of the probability of death to survival. When a bycatch occurs with a constant probability (instantaneous hazard rate, $m$), the risk of bycatch $S(t)$ at total presence duration $t$ is represented as: $S(t) = 1 - \exp(-mt)$. This equation means that the longer porpoises are present near a net, the more vulnerable to entanglement they become. Although the duration per presence had not a significant difference, the total duration per hour was obviously different because the number of presences was largely different between BB and NB. However, it is difficult to estimate $m$ due to the small number of present samples, though further research may enable the risk assessment of bycatch.

Previous studies have investigated the reaction of harbour porpoises to nets, demonstrating that both captive [48] and wild harbour porpoises [34] will avoid nets in a short distance (<100 m). In contrast, the current study shows active foraging activity by harbour porpoises around a bottom-gillnet, while the detection range may be larger than 100m. The behaviour of porpoises is best observed under actual circumstances where bycatches are occurring because environmental factors such as prey availability affect the seasonal and diel occurrence of cetaceans [45]. Additional studies at actual fishery grounds are needed to determine the factors governing the incidence of bycatch. Furthermore, it is important to recognise the environmental conditions that affect the behaviour of porpoises, particularly prey distributions and movements.

The main conclusion of this study is that the harbour porpoises frequently and repeatedly appeared around a bottom-gillnet set-up in a commercial fishing ground, and one reason for the frequent occurrence was probably that forage fish (*Ammodytes* sp.) were distributed in the fishing ground. Because the *Ammodytes* sp. themselves were not captured by the net, it is clear that the porpoises' foraging activity was not depredation of the net.

Some mitigation measures have been proposed for porpoise bycatch: time-area restrictions on fishing effort [15] corresponding to the density of *Ammodytes* sp., technological modifications to fishing gear [49], and the use of acoustic alarms known as pingers [35]. However, it has been recognised that odontocetes are likely to habituate with acoustic deterrent devices if coupled with the presence of food [50]. Thus, for more efficient reductions of bycatch, these measures could be used in combination, such as deploying acoustic alarms coupled with area closure [23], while improving the acoustic reflectivity of fishery nets may be another efficient way to increase detectability by the porpoises' biosonar.

## Materials & methods

### Study location and bottom-gillnet set-up

Observations were carried out at a bottom-gillnet set located off Rausu, Hokkaido, Japan (43˚ 52.52'N, 145˚09.90'E), in July and August of 2016 to 2018 (Table 1). The observation point was located 3 nautical miles (5.6 km) from the shoreline, at a bottom depth of 50–55 m, on a rocky seafloor. The gillnet fishing and the acoustic observations were permitted under the fishing permission of a fisher (K. S.) belonging to Rausu Fishing Cooperative Association and a co-author of this paper in the fishing ground with the common fishery right permitted by the governor of Hokkaido to the Rausu Fishing Cooperative Association.

The observed bottom-gillnet was ~1 900 m in length, with 7.5 cm-mesh. Both ends of the net were fixed by weights positioned approximately 1 200 m apart (Fig 1). The net meandered along the seafloor and did not stand up vertically. Fishing operations were carried out once a day, from 01:30 to 03:00, except for days with rough sea conditions. During the fishing operation, the net set on the seafloor was retrieved and a replacement bottom-gillnet was deployed at approximately the same site. The operation of this bottom-gillnet had been carried out with a cycle of 24 h, if a porpoise became entangled, the individual was landed.

Bottom-gillnets are used mainly to target a variety of righteye flounders (Pleuronectidae): dusky sole (*Lepidopsetta mochigarei*), starry flounder (*Platichthys stellatus*), black plaice (*Pseudopleuronectes obscurus*), yellow striped flounder (*Pseudopleuronectes herzensteini*), sand flounder (*Pleuronectes punctatissimus*), pointhead flounder (*Cleisthenes pinetorum*), willowy flounder (*Glyptocephalus kitaharai*), and Pacific halibut (*Hippoglossus stenolepis*). Sculpins (Cottidae spp.) are also frequently caught, although not targeted: antlered sculpin (*Enophrys diceraus*), great sculpin (*Myoxocephalus polyacanthocephalus*), plain sculpin (*Myoxocephalus jaok*), and sea raven (*Hemitripterus villosus*). In addition, mottled skate (*Raja pulchra*), Japanese amberjack (*Seriola quinqueradiata*), and chestnut octopus (*Octopus conispadiceus*) are occasionally caught.

The bycatch of harbour porpoises occurred three times during the observations (Tables 1 and 2). All of the bycatch individuals were collected during normal fishing operations. The period before bycatch (BB) was defined as the period from when the gillnet was set up to when a porpoise became entangled (i.e. from the end of the fishing operation on the previous day to the commencement of the fishing operation during which the bycatch occurred). Periods apart from BB were defined as 'no bycatch' (NB).

## Passive acoustic monitoring

The A-tag (Fixed Type; Marine Micro Technology, Saitama, Japan) is an ultrasonic pulse event recorder that stores received sound-pressure level, sound arrival-time differences between two hydrophones, and an inter-click interval (ICI), which is the interval between the envelope peaks of consecutive clicks, when the sound is higher than the pre-set threshold level (138 dB peak-to-peak re 1/µPa). The sampling frequency was 2 kHz, which gives a time resolution of pulse event detection of 0.5 ms. Incoming signals were bandpass filtered (55–235 kHz), which included the peak frequency of harbour porpoise clicks (129–145 kHz) [51]. The two hydrophones were fixed to an aluminium bar, 65 cm apart, to record the time difference of each pulse, which provided the elevation/depression angle direction of the recorded sound. An A-tag can record for one month. The maximum detectable distance of the A-tags was estimated to be about 750 m [40], but most of the detected distance will be much shorter (several 100's m) than the maximum distance because of the directivity of the sonar and the masking by the noise.

The deployment of the A-tags and the bottom-gillnet is illustrated in Fig 1. A-tags were independently fixed near both ends of the gillnet. Distance between the two tags was about 1 200 m, and the tags were set about midway to the bottom depth (at ~25 m above the seafloor), where wave turbulence and swell were limited. The A-tags were vertically placed in relation to the two hydrophones. Because of the set-up, elevation/depression angle indicates the depth layer of harbour porpoise (Fig 7). If the elevation/depression angle is a negative value, it means that the harbour porpoise appeared deeper than the A-tag (i.e. below 25-m depth).

All recorded clicks were assumed to be emitted from harbour porpoises. A-tag is tuned to detect the dominant frequency of the clicks of harbour porpoises. Although Dall's porpoise may emit similar clicks and might use this area [52], the most frequently (more than 95%)

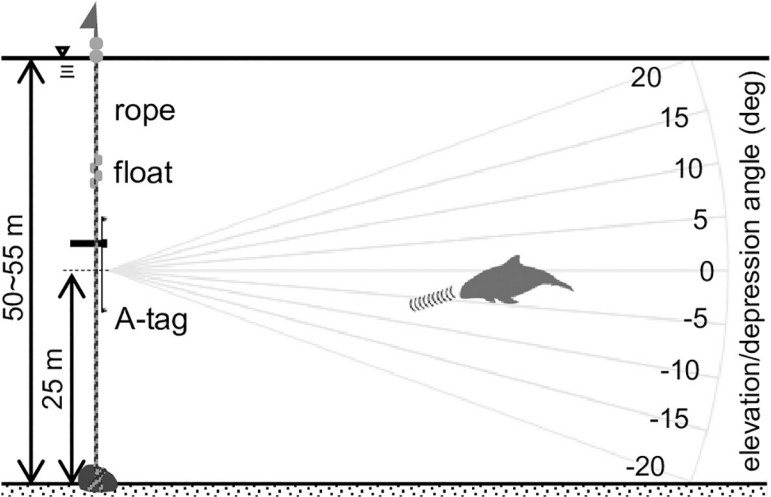

**Fig 7. Diagram of the A-tag deployment and the relationship between recorded elevation/depression angle and the tag in the gillnet set-up off Rausu, Hokkaido, Japan.** Numbers around the arc indicate elevation/depression angle (degrees).

observed toothed whale and the only toothed whale that incidentally caught by this bottom-gillnet was harbour porpoise.

## Off-line click selection

An off-line filter using Igor Pro 6.2 (Wave Metrics, Lake Oswego, Oregon, USA) was used for data collection to exclude background noises and pseudo pulse sounds. First, the data that was not triggered by both hydrophones were excluded. Successive pulses within 3 ms were omitted, assuming these to be a reflection of sound from the water surface or seabed. The remaining pulse sounds were divided into groups termed click-trains, which were defined as groups of sequential pulses separated from other pulses by an ICI >200 ms, comprising ≥6 pulses [38]. The pulses with an elevation/depression angle between 2.5˚ and 3.5˚ were assumed to be noises propagated by interference with the clock frequency of CPU of the A-tag, and thus were removed for the analysis. Finally, manual checks were conducted to exclude contamination noises caused by large ships or artificial sonar sounds, and to select the pulses to be used in further analysis.

## Calculation of harbour porpoise presence

To avoid multiple counting from a single presence of a porpoise, the presence threshold interval (PTI) was defined as the value at 95% cumulative frequency of the interval between click-trains [30, 53]. In this study, PTI was determined as 650 sec (Fig 8). When a click-train sequence separated by a click-train interval greater than PTI was observed, it was determined to denote 'presence' of harbour porpoise. Furthermore, the 'presence duration' and 'presence probability' were measured to assess the characteristics of each presence. The presence duration was defined as the length of time that the presence lasted. The presence probability was defined as the ratio of total presence duration hour$^{-1}$.

## Classification of foraging sequences

Porpoises emit specific click patterns when capturing prey [40]. A click-train sequence comprises different phases by changes in ICI, approach phase (initial and terminal part), to be

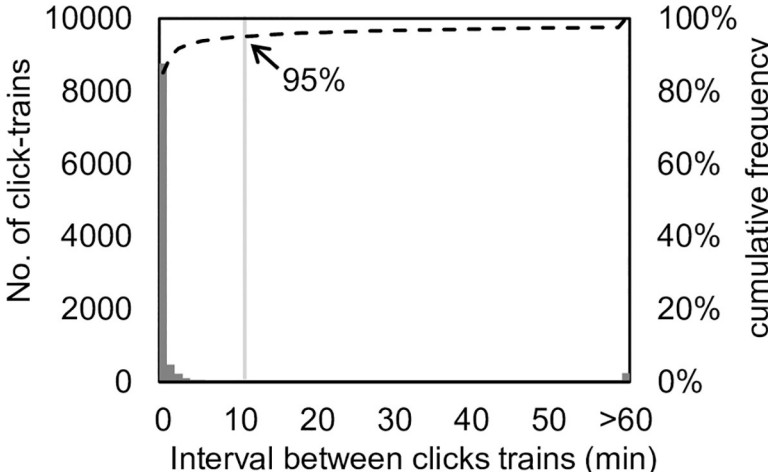

**Fig 8. Frequency distribution of recorded intervals between click-trains of harbour porpoises recorded by the A-tags.** The black bars show the number of click-trains at the corresponding interval between clicks-trains; the dashed line shows the cumulative distribution of the frequencies.

defined as a possible prey-capture sequence or foraging sequences. The initial part of the approach phase is characterised by a relatively stable ICI of about 50 ms [39]. The terminal part of the approach phase is marked by a sudden and rapid decrease of ICI, below 10 ms. Previous studies defined a foraging sequence based on passive monitoring in the wild [37, 41, 51]. Following these studies, the criterion for foraging sequence was defined as a click-train sequence that included a sudden decrease of ICI below 30 ms. The clicks with zero elevation/depression angle were excluded from the foraging sequences, because these indicated that a porpoise was approaching an A-tag horizontally, thus the porpoise was presumably not foraging but likely just scrutinising the A-tag.

## Statistical analysis

The average number of foraging sequences per hour were compared between daytime (5:00–19:00) and night-time (19:00–5:00) by using *t*-test. Also, the average of the behavioural parameters (number of presences, presence duration, presence probability, elevation/depression angle, and number of foraging sequences) were compared between BB and NB by using *t*-test.

## Stomach-contents analysis of bycatch individuals

Stomachs were collected from 2 individuals, bycaught by the bottom-gillnet in 2017 (Table 2). The porpoises were necropsied, and their stomachs excised and frozen for later examination in the laboratory. The stomach contents were preserved in 80% ethanol for sorting of the lower beaks of cephalopods and the otoliths of fish. The lower beaks and otoliths were counted and identified to the lowest possible taxonomic level, by referring to published guides [54–56]. Numerical composition of each prey category ($N_i$%) was determined; $N_i$% indicates a numerical percentage of the *i*-th prey item in relation to the total number of prey individuals found in the stomachs.

## Acknowledgments

Cynthia Kulongowski (MSc), with the Edanz Group (www.edanzediting.com/ac), edited a draft of this manuscript.

## Author Contributions

**Conceptualization:** Saki Maeda, Tomonari Akamatsu, Takashi Fritz Matsuishi.

**Data curation:** Saki Maeda, Kenji Sakurai, Tomonari Akamatsu, Ayaka Matsuda, Orio Yamamura, Mari Kobayashi, Takashi Fritz Matsuishi.

**Formal analysis:** Saki Maeda.

**Funding acquisition:** Ayaka Matsuda, Takashi Fritz Matsuishi.

**Investigation:** Saki Maeda, Kenji Sakurai, Ayaka Matsuda, Mari Kobayashi.

**Methodology:** Saki Maeda, Kenji Sakurai, Takashi Fritz Matsuishi.

**Project administration:** Takashi Fritz Matsuishi.

**Supervision:** Tomonari Akamatsu, Takashi Fritz Matsuishi.

**Validation:** Saki Maeda, Takashi Fritz Matsuishi.

**Visualization:** Saki Maeda.

**Writing – original draft:** Saki Maeda.

**Writing – review & editing:** Tomonari Akamatsu, Ayaka Matsuda, Orio Yamamura, Mari Kobayashi, Takashi Fritz Matsuishi.

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
