## [Decision Letter · Decision Letter 0]

11 Dec 2020

PONE-D-20-26922

Foraging activity of harbour porpoises around a bottom-gillnet in a coastal fishing ground, under the risk of bycatch

PLOS ONE

Dear Dr. MATSUISHI,

Thank you for submitting your manuscript to PLOS ONE. After careful consideration, we feel that it has merit but does not fully meet PLOS ONE’s publication criteria as it currently stands. Therefore, we invite you to submit a revised version of the manuscript that addresses the points raised during the review process.

First, I must apologize for the delay in your manuscript. Two reviewers agreed to review the manuscript, but never submitted their reviews, which added significant delays to the review process as I had to secure entirely new reviews after waiting.

Your manuscript has been assessed by two experts in the field. You'll see that Reviewer 1 is quite critical, but they do have relevant points. I do not agree with this reviewer that 3 cases of bycatch is too small a sample size for this manuscript. However, in response to this critique, the authors must ensure that the manuscript is framed around this small sample size and not make generalizations about porpoise bycatch in general, but rather make it clear that this study is based on a small sample and the implications of this small sample.

Reviewer 2 had a number of very constructive comments that must be addressed when revising the manuscript. In particular, the lack of proper reporting of statistical tests is a major issue, and this information must be added in.

We look forward to receiving your revised manuscript.

Kind regards,

William David Halliday, Ph.D.

Academic Editor

PLOS ONE

Journal Requirements:

Reviewers' comments:

Reviewer's Responses to Questions

**Comments to the Author**

1. Is the manuscript technically sound, and do the data support the conclusions?

Reviewer #1: Partly

Reviewer #2: Yes

2. Has the statistical analysis been performed appropriately and rigorously? 

Reviewer #1: No

Reviewer #2: No

3. Have the authors made all data underlying the findings in their manuscript fully available?

Reviewer #1: No

Reviewer #2: No

4. Is the manuscript presented in an intelligible fashion and written in standard English?

Reviewer #1: Yes

Reviewer #2: Yes

5. Review Comments to the Author

Reviewer #1: This manuscript includes some important results which could contribute to understanding the mechanism of bycatch of harbor porpoises. However, I do not recommend accepting the manuscript due to several critical drawbacks. The most serious concern is insufficient data of bycatch cases. They compared porpoise presences between bycatch and non-bycatch periods only in three cases. I do not think this is enough to derive a reasonable conclusion.

The authors failed to indicate how their results contribute to revealing the mechanism of bycatch and its mitigation. The insufficient results seem to hamper this.

Another concern is about the definition ‘around the net’. The distance should also be a crucial factor in bycatch. I think that 730m of A-tag detection range from both ends of the net is too large to say that the porpoise approached the net itself. The ‘presence of the porpoise’ means that the porpoise is somewhere 0m and 730m from the net. The bycatch possibility should be quite different by the distance from the net, but this is not considered in this setting. If A-tag has a horizontal directivity toward the net, this is not the case.

The arguments on avoidance of the net also include a similar problem. The control setting is required to know there is no avoidance of the net. Thus, the current results are not comparable to those of the previous studies that showed avoidance of the net.

It is pretty natural the longer stay around the net results in a higher risk of bycatch. However, not the duration but the number of presence and probability were different between BB and NB. This is not fully discussed.

Reviewer #2: This study details passive acoustic monitoring of porpoises around an operational fishery in Hokkaido, Japan, to determine behaviour around fishing nets. The authors document frequent acoustic detections of porpoises around the net, including foraging buzzes. Three bycatch events were recorded and these were associated with increased porpoise detections, suggesting entanglement results from time spent foraging in the vicinity of nets. As we still have a relatively poor understanding of how porpoises become entangled, this study adds important information that could be useful for conservation.

General comments:

I enjoyed reading the study and commend the authors for its simplicity. However, I think it lacks the relevant information in places. In particular, it appears that there is no information about or justification for the statistical tests used in the Methods section, and so I cannot provide an assessment of their appropriateness. Also, in the discussion, there is scope for the authors to more explicitly detail how these results could be used to manage porpoise-fishery interactions, especially in the context of the bottom-set gillnet fishery in Japan.

One of the strengths of the study is that by using an A-tag at each end of the net, the whole area of the net was covered acoustically (i.e. in theory there are no blind spots). However, a potential limitation with this set-up is that the recording ranges of the two loggers might have overlapped slightly (given the detection ranges of both combined are greater than the length of the net) which could result in pseudo-replication (i.e. double recording of the same individuals), especially if data from both A-tags were pooled. Regardless, there is no mention in the paper about how data from separate A-tags are dealt with. This information is important as the appropriateness of statistical tests will depend on the ways in which the datasets were first treated.

Specific comments:

L26: it’s worth mentioning that these species were not targeted directly by the fishery (if true).

L30-32: would it be possible to provide some management recommendations from this work, perhaps in a sentence here?

L36-37: As the first page of the introduction is broad, presenting generalities of small cetacean bycatch before focussing on harbour porpoises, it seems odd to mention gillnets in Japan in the second sentence. Perhaps you can move this text elsewhere?

L38: change to “bycatch in gillnets”.

L44-45: although this is not necessarily true as in many parts of the world such as South America (e.g. Campbell et al. 2020) or West Africa (e.g. Leeney et al. 2015), small cetaceans are either directly targeted for consumption or for use as bait in gillnet and longline fisheries (reviewed in Mintzer et al. 2018). Therefore, cetacean bycatch often does have both indirect and direct (still traded illegally at fish markets) economic value, and so cannot always be considered “undesirable” for fishers.

L45: what do you mean by “fishery efficiency” – is it profitability? See my previous point that this may not always be the case. Crucially also, reducing cetacean bycatch is important for fisheries sustainability and the viability and persistence of cetacean populations.

L54-57: although studies using pingers in operational fisheries have found no evidence of increased bycatch over time (Palka et al. 2008, Carretta and Barlow 2011).

L57: reference Kindt-Larsen et al. 2019 as this study deals with some of factors affecting potential habituation (e.g. pinger type, duty-cycling).

L61 & L69: do you have references for these observations?

L73-76: Read et al. 2003 is a good reference showing bottlenose dolphins depredate fish from nets (and are often not bycaught).

L85-86: could you say specifically how you expect reactions to differ? You could argue that increased foraging in the vicinity of nets increases likelihood of bycatch, but alternatively if animals are travelling (and so not vocalizing as frequently as during foraging), they may not detect nets and so entanglement risk may be higher.

L87-92: however, a recent study of the similarly-related Burmeister’s porpoise used a similar approach in gillnet fisheries in Peru and found acoustic detections were higher in sets with porpoise bycatch, showing that increased time spent around nets increases bycatch risk (Clay et al. 2018).

L104: could you add more specific aims here? For example you look at whether foraging activity varies by day or night, by depth and you link acoustic detections to bycatch events. Some predictions as to what you expect (based on what’s written in the introduction) would be useful.

L109: as the results section appears to come before the methods, it would be good to briefly summarize what you did here (i.e. loggers attached to nets for and recorded acoustic activity for .. days).

L110: how many days were nets observed?

L149: do you have any idea of when mortality occurs? For example, is there any evidence of a sudden reduction in acoustic detections that might be apparent when the animal drowns in the net (and dies), that might be detected by both A-tags? I presume that if there are other animals vocalizing around the net, this may not be possible…

L179: it would be good to see a plot of detections over time for the study period, not just according to the diel cycle.

L194-195: are these the predominant prey species of harbour porpoise based on previous studies in the region or elsewhere (with a greater sample size than n=3 individuals)?

L195: how frequently were they observed – is it possible to incorporate these observations into the study in a more quantitative way?

L209-210: do any of the species targeted by the gillnet fishery feed on Ammodytes sp.?

L215: a similar conclusion was made in a study linking detections of the Burmeister’s porpoise to bycatch events in the Peruvian drift-net fishery (Clay et al. 2018), so I suggest mentioning this study here.

L225-227: the detection range from the cited study showed aversive reactions at <100 m however your quoted detection range is up to 750 m, which suggests porpoises could avoid nets at distances of up to several 100s m and still be detected acoustically within a larger radius. I suggest rephrasing the text slightly to emphasize that animals could be detected foraging near the net but be them over small distances.

L233-237: given these conclusions, can you make any recommendations about how the fishery can reduce bycatch? For example, could fishing activity be concentrated in areas of high densities of target prey relative to Ammodytes species (and presumably porpoises)?

L240-241: research shows that initial responses to pingers of neophobic species such as porpoises are aversive, but that this response decrease over time (i.e. habituation). So, I am not sure you can conclude that they “tolerate” pingers - I suggest a slight rephrase.

L238-244: similar to my point above, the paper would really benefit from a consideration of which of the mitigation measures listed are actually feasible in this fishery. For example, which measure (or combination) would gain traction among local fishers?

Methods:

This section is lacking any detail regarding choice of statistical tests. For example, what tests do you use to compare day vs night patterns, foraging by depth, and link acoustic detections to bycatch days (and controlling for extremely small sample size in BB vs NB periods)? Also, how do you define day and night?

L248: how many days during this period were monitored?

L276: change “soaked” to “set”.

L303-305: are there any other narrow-band high frequency (NBHF) in the region? From other studies (e.g. Ohizumi et al. 2000), it seems that Dall’s porpoise may also use the area during summer.

Table 4 legend: change “was” to “were”.

Fig. 2 & 5: it would be good to combine these two figures, and perhaps have a third panel showing proportion of total detections that are foraging buzzes, to demonstrate how foraging propensity changes with depth.

References

Campbell E, Pasara-Polack A, Mangel JC, Alfaro-Shigueto J (2020) Use of Small Cetaceans as Bait in Small-Scale Fisheries in Peru. Front Mar Sci 7.

Carretta JV, Barlow J (2011) Long-Term Effectiveness, Failure Rates, and “Dinner Bell” Properties of Acoustic Pingers in a Gillnet Fishery. Mar Technol Soc J 45:7–19.

Clay TA, Mangel JC, Alfaro-Shigueto J, Hodgson DJ, Godley BJ (2018) Distribution and Habitat Use of a Cryptic Small Cetacean, the Burmeister’s Porpoise, Monitored From a Small-Scale Fishery Platform. Front Mar Sci 5:220.

Kindt‐Larsen L, Berg CW, Northridge S, Larsen F (2018) Harbor porpoise (Phocoena phocoena) reactions to pingers. Mar Mammal Sci.

Leeney RH, Dia IM, Dia M (2015) Food, Pharmacy, Friend? Bycatch, Direct Take and Consumption of Dolphins in West Africa. Hum Ecol 43:105–118.

Mangel JC, Alfaro-Shigueto J, Van Waerebeek K, Cáceres C, Bearhop S, Witt MJ, Godley BJ (2010) Small cetacean captures in Peruvian artisanal fisheries: High despite protective legislation. Biol Conserv 143:136–143.

Mintzer VJ, Diniz K, Frazer TK (2018) The Use of Aquatic Mammals for Bait in Global Fisheries. Front Mar Sci 5.

Ohizumi H, Kuramochi T, Amano M, Miyazaki N (2000) Prey switching of Dall’s porpoise Phocoenoides dalli with population decline of Japanese pilchard Sardinops melanostictus around Hokkaido, Japan. Mar Ecol Prog Ser 200:265–275.

Palka DL, Rossman MC, Vanatten AS, Orphanides CD (2008) Effect of pingers on harbour porpoise (Phocoena phocoena) bycatch in the US Northeast gillnet fishery. J Cetacean Res Manag 10:217–226.

Read AJ, Waples DM, Urian KW, Swanner D (2003) Fine-scale behaviour of bottlenose dolphins around gillnets. Proc Biol Sci 270 Suppl 1:S90-2.

6. PLOS authors have the option to publish the peer review history of their article (what does this mean?). If published, this will include your full peer review and any attached files.

Reviewer #1: No

Reviewer #2: No

---

## [Author Response · Author response to Decision Letter 0]

14 Jan 2021

Please see attached "Response to Reviewers R1.docx"

---

## [Decision Letter · Decision Letter 1]

25 Jan 2021

PONE-D-20-26922R1

Foraging activity of harbour porpoises around a bottom-gillnet in a coastal fishing ground, under the risk of bycatch

PLOS ONE

Dear Dr. MATSUISHI,

Thank you for submitting your manuscript to PLOS ONE. After careful consideration, we feel that it has merit but does not fully meet PLOS ONE’s publication criteria as it currently stands. Therefore, we invite you to submit a revised version of the manuscript that addresses the points raised during the review process.

One reviewer from the previous round and I reviewed the manuscript, and we find it greatly improved. Only some minor revisions are required before it can be acceptable. Please address all comments below, as well as my embedded comments in the uploaded file.

We look forward to receiving your revised manuscript.

Kind regards,

William David Halliday, Ph.D.

Academic Editor

PLOS ONE

Additional Editor Comments (if provided):

This manuscript is very close to being acceptable. The reviewer thought all major issues had been dealt with, and just provided some very minor comments. I agree with this assessment. I have read through the manuscript in detail, and provide a number of basic formatting and editorial comments in the uploaded document.

Paragraph structure is poor throughout, with many short paragraphs that should likely be combined. A paragraph should have a singular theme, and there are very clear examples of small paragraphs in the introduction that should be combined together, which I highlight in the attached document. Please apply this advice throughout the manuscript.

The methods section needs to be moved up in the manuscript before the Results.

Reviewers' comments:

Reviewer's Responses to Questions

**Comments to the Author**

1. If the authors have adequately addressed your comments raised in a previous round of review and you feel that this manuscript is now acceptable for publication, you may indicate that here to bypass the “Comments to the Author” section, enter your conflict of interest statement in the “Confidential to Editor” section, and submit your "Accept" recommendation.

Reviewer #2: All comments have been addressed

2. Is the manuscript technically sound, and do the data support the conclusions?

Reviewer #2: Yes

3. Has the statistical analysis been performed appropriately and rigorously? 

Reviewer #2: Yes

4. Have the authors made all data underlying the findings in their manuscript fully available?

Reviewer #2: No

5. Is the manuscript presented in an intelligible fashion and written in standard English?

Reviewer #2: Yes

6. Review Comments to the Author

Reviewer #2: I have been through the revised MS and can confirm that the authors have done a great job dealing with my comments, and the MS have improved as a result. I congratulate the authors on their valuable contribution. I have some very minor comments (relating mostly to grammar/spelling) that will help tighten the language further:

L33 (line numbers in cleaned version): change “were” to “are”.

L37: “gillnets”.

L114: change “material part” to “Materials and Methods section”.

L124: is this stated previously? Maybe should be “as stated below” if referring to methods?

L126: insert “negative” before “value”.

L182: would be good again here to reiterate what “period category BB” is in real terms to make it clearer to the reader – i.e. on the day prior to a set with a bycatch event…vs days prior to sets with no bycatch event?

L251: change to “combination”.

7. PLOS authors have the option to publish the peer review history of their article (what does this mean?). If published, this will include your full peer review and any attached files.

Reviewer #2: No

---

## [Author Response · Author response to Decision Letter 1]

25 Jan 2021

See the Point-by-point responses file.

---

## [Editor Report · Decision Letter 2]

27 Jan 2021

Foraging activity of harbour porpoises around a bottom-gillnet in a coastal fishing ground, under the risk of bycatch

PONE-D-20-26922R2

Dear Dr. MATSUISHI,

We’re pleased to inform you that your manuscript has been judged scientifically suitable for publication and will be formally accepted for publication once it meets all outstanding technical requirements.

Kind regards,

William David Halliday, Ph.D.

Academic Editor

PLOS ONE
---

## [Editor Report · Acceptance letter]

2 Feb 2021

PONE-D-20-26922R2 

Foraging activity of harbour porpoises around a bottom-gillnet in a coastal fishing ground, under the risk of bycatch 

Dear Dr. Matsuishi:

I'm pleased to inform you that your manuscript has been deemed suitable for publication in PLOS ONE. Congratulations! Your manuscript is now with our production department. 

Kind regards, 

on behalf of

Dr. William David Halliday 

Academic Editor

PLOS ONE